# A Revision of Herpes Simplex Virus Type 1 Transcription: First, Repress; Then, Express

**DOI:** 10.3390/microorganisms12020262

**Published:** 2024-01-26

**Authors:** Laura E. M. Dunn, Claire H. Birkenheuer, Joel D. Baines

**Affiliations:** Baker Institute for Animal Health, Department of Microbiology and Immunology, Cornell University, Ithaca, NY 14850, USA; led97@cornell.edu (L.E.M.D.); chb226@cornell.edu (C.H.B.)

**Keywords:** transcription, ICP4, ICP0, RNA polymerase II, ICP22, HSV-1, herpes simplex virus

## Abstract

The herpes virus genome bears more than 80 strong transcriptional promoters. Upon entry into the host cell nucleus, these genes are transcribed in an orderly manner, producing five immediate–early (IE) gene products, including ICP0, ICP4, and ICP22, while non-IE genes are mostly silent. The IE gene products are necessary for the transcription of temporal classes following sequentially as early, leaky late, and true late. A recent analysis using precision nuclear run-on followed by deep sequencing (PRO-seq) has revealed an important step preceding all HSV-1 transcription. Specifically, the immediate–early proteins ICP4 and ICP0 enter the cell with the incoming genome to help preclude the nascent antisense, intergenic, and sense transcription of all viral genes. VP16, which is also delivered into the nucleus upon entry, almost immediately reverses this repression on IE genes. The resulting de novo expression of ICP4 and ICP22 further repress antisense, intergenic, and early and late viral gene transcription through different mechanisms before the sequential de-repression of these gene classes later in infection. This early repression, termed transient immediate–early protein-mediated repression (TIEMR), precludes unproductive, antisense, intergenic, and late gene transcription early in infection to ensure the efficient and orderly progression of the viral cascade.

## 1. Introduction

Infection with herpes simplex virus type 1 (HSV-1) leads to lifelong infection, with an estimated 67% of adults worldwide infected [1]. HSV-1 persists in a biphasic lifecycle, in which it switches between lytic and latent transcriptional programs. The latent phase occurs in sensory neurons, where the viral genome is maintained as an episome and gene expression is limited to production of the latency-associated transcript (LAT), whereas, in the lytic phase, primarily in epithelial cells, over 80 genes are expressed in a highly regulated temporal cascade. This remarkable ability to regulate the transcription of two distinct phases is key to establishing and maintaining persistent infection. To achieve this, HSV-1 repurposes cellular RNA polymerase II (Pol II) for viral transcription. 

Recent technical advances in cryo-electron microscopy (cryo-EM) have greatly enhanced the understanding of cellular transcriptional regulation [2,3,4,5,6,7]. These studies have revealed detailed structure/function relationships within and between different complexes. These interactions initiate transcription from an initiation complex, facilitate Pol II promoter clearance and RNA capping, and induce proximal pausing. These steps are followed sequentially by pause release, elongation, nascent RNA cleavage, and termination [8], which are discussed in more detail below. It is useful to consider this new information and what it may mean for understanding HSV transcription, especially considering new deep sequencing analyses of nascent RNA arising from the HSV-1 genome.

## 2. Overview of RNA Polymerase II-Mediated Transcription

The steps of eukaryotic transcription are summarized in Figure 1.

### 2.1. Formation of the Pre-Initiation Complex

Gene activation requires the formation of a pre-initiation complex (PIC) at the promoter. The first step in PIC formation in the classical model involves the TATA-binding protein (TBP) binding to promoter-region DNA [9,10,11]. This step is facilitated by the remodeling/removal of promoter-associated nucleosomes. The acetyltransferase paralog p300 and the CREB binding protein (p300/CBP) provide some of this remodeling [12]. TBP-bound DNA can associate other TBP-associated factors (TAFs), forming the larger TFIID general transcription factor [9]. TFIID binding to promoters is facilitated by specific histone marks within promoter-bound and promoter-proximal nucleosomes [13].

TBP/TFIID binding physically bends the DNA by about 90°, which allows for interaction with TFIIB [14]. TFIIB binds the RPB1 subunit of Pol II, which subsequently co-recruits the TFIIF complex as it interacts with the RPB2 subunit of Pol II [11]. Together, these components induce structural changes within the 12 sub-unit Pol II, anchoring the DNA along the enzyme and positioning the complex correctly over the transcriptional start site (TSS) [11,15].

A large complex, termed Mediator, recognizes the properly formed PIC and essentially covers Pol II and bound TFII general transcription factors [11,16,17]. It also binds to promoter-specific transcription factors located upstream from the TFII general transcription factors [17,18]. Mediator coordinates signals from the promoter-specific transcription factors to the PIC, inducing the recruitment of the TFIIH and TFIIE general transcription factors [19].

### 2.2. Transcription Initiation

Together, TFIIH and TFIIE open the promoter DNA, creating a transcription bubble and trapping the coding strand of DNA into the Pol II cleft [15]. Once the promoter is open, Mediator positions the cyclin-dependent kinase (CDK)-activating kinase domain (CAK) of TFIIH over the C-terminal domain (CTD) of RPB1, where it can phosphorylate Ser5 in the heptad repeat of the RPB1 CTD [16,20,21,22]. CDK7 kinase activity is associated with DNA opening and Pol II clamp closure over the template DNA, which starts transcription [11,15,23]. CDK7 activity on Ser5 in the CTD is also required to recruit RNA capping enzymes to the complex [24,25]. The structural rearrangements induced by the TFIIE, TFIIH, and CDK7 phosphorylation of the Pol II cause the initiation of transcription and the synthesis of nascent RNA [11,16,26].

### 2.3. Promoter Clearance and Nascent RNA Capping

The complete process of initiation also requires structural rearrangements that occur after the nascent RNA reaches a length of 12–13 nt [11,27,28]. These rearrangements include (i) TFIIB release from the Pol II nascent RNA exit channel, (ii) the recruitment of the nascent RNA capping machinery, and (iii) the disassociation of the initiated Pol II complex from the PIC [11,26].

The nascent RNA capping machinery (MCE1) binds to the transcriptional complex at a time when only Ser5 in the CTD of RPB1 is phosphorylated [29]. This pattern of phosphorylation only occurs when the Pol II first initiates transcription and before it has cleared the promoter [30]. The resulting nascent RNA tail exiting Pol II is engaged by the capping machinery around 19–22 nt downstream of the transcription start site [31]. Further phosphorylation events induced by CDK7 in TFIIH release Pol II from Mediator and the general transcription factors [26,32].

### 2.4. Promoter Proximal Pausing

Promoter proximal pausing (PPP) is a highly conserved feature of many bacterial and eukaryotic genes, and in eukaryotes, it is controlled in part by the DRB-sensitive inducing factor (DSIF), the negative elongation factor (NELF), and the positive elongation factor complex (P-TEFb) that contains cyclin-dependent kinase 9 (CDK9) [33,34,35,36]. After promoter clearance and 50 to 500 nt of nascent RNA is produced, eukaryotic Pol II is inhibited by DSIF and NELF [5,37,38]. In the paused state, the Spt4 and Spt5 subunits of DSIF lie over the RPB1 and RPB2 subunits of Pol II [2,7,39], and occupy similar binding sites to the TFIIB, TFIIE, and TFIIF general transcription initiation factors [39,40,41]. Some of these components are left behind at the gene promoter once transcription initiation commences [42,43]. It is thought that one major reason for PPP at this location is to allow for the completion of RNA capping [44,45].

Unphosphorylated DSIF binding to the RPB2 subunit causes internal confirmational changes within Pol II. These induced changes switch the physical structure of the enzyme into its elongating form, which can transcribe at a faster rate than the initiating form [2,7,39,46,47]. Unphosphorylated DSIF also forms a DNA/RNA clamp over the respective exit channels in the RPB2 subunit of Pol II [2,5,6]. By binding to the upstream DNA coming out of the Pol after it has been transcribed, the nascent RNA emerging from the RNA exit tunnel with the RPB2 subunit and unphosphorylated DSIF “tether” the Pol II to the previously transcribed DNA, preventing additional nascent RNA production. These events physically stop the process of transcription near the promoter of the gene. NELF then (i) binds the Pol II/DSIF complex and blocks nucleotides from entering the active site of Pol II, (ii) prevents the binding of important transcriptional elongation factors [6,48], and (iii) stabilizes the paused Pol II so that it does not terminate prematurely [49].

### 2.5. Release of Promoter Proximal Pausing

The P-TEFb complex then “relaxes” the paused state after the correct nascent RNA processing enzymes are assembled. The main signal to induce this conformational change is through the CDK9 phosphorylation of multiple transcriptional targets. These targets include NELF, DSIF, and Ser2 in the heptad repeat of RPB1 [50]. The CDK9 phosphorylation of NELF causes its disassociation from the transcriptional complex, allowing nucleotide entry [35,51,52,53]. Phosphorylated DSIF becomes a positive elongation factor, releasing its DNA/RNA clamps and “riding” the elongating Pol II through the gene body [54,55,56]. Phosphorylated DSIF also recruits the transcription elongation/chromatin remodeling factor, Spt6 [6]. Lastly, CDK9 phosphorylates Ser2 in the heptad repeat of RPB1’s CTD. The CDK9 phosphorylation of RPB1’s CTD correlates with PPP release, yet this phosphorylation event only indirectly triggers the release [36,43,57]. Instead, this modification more likely enhances the interaction of Pol II with the Spt6 histone chaperone complex, as well as other transcription elongation factors [43,58,59].

### 2.6. Elongation

During elongation, Spt6 and FACT act as histone chaperones; the FACT complex disassembles histones ahead of Pol II, and Spt6 helps to reassemble them downstream after Pol passes [4,6,60]. Unlike Spt6, which is recruited to genes in part via Ser2 phosphorylation in the heptad repeat of RPB1’s CTD, FACT is recruited to the transcriptional complex by the +1 nucleosome interacting with the chromatin remodeler Chd1 [4,6,43]. The phosphorylated DSIF, the FACT complex, and Spt6 all control the rate of transcription through the gene body, using various mechanisms [4,7,39]. This is necessary because the rate of Pol II transcription elongation must be tightly controlled to ensure the proper processing of the nascent RNA [46,61,62].

### 2.7. Termination

Finally, Spt6 interacts with the Aly/Ref component of the TREX complex though its association with the IWS1 (Interaction with Spt6/Spn1) protein [63,64]. This interaction facilitates cross-talk between the elongating Pol, histone remodeling factors, and the nascent RNA emerging from the exit channel, and, in turn, is important in mRNA processing and export to the cytoplasm [63].

## 3. HSV-1 Transcription

During productive HSV-1 infection, viral genes are expressed in distinct temporal stages, beginning with immediate–early (IE) genes, which are followed sequentially by early (E), leaky late (LL), and true late (L) genes [65,66]. There are five IE genes (α0, α4, α22/US1, α27/UL54, and α47/US12) and all share a TAATGARAT motif in their promoters, which is required for their expression [67]. VP16, a virion component that enters the nucleus upon entry, is a potent TAF which recognizes and binds TAATGARAT alongside the cellular transcription factors HCF and Oct-1 [68,69], leading to robust IE gene transcription within minutes of infection [70].

E and LL/L gene promoters do not have the TAATGARAT motif and therefore do not utilize VP16 to initiate expression. Instead, these genes require another virus-encoded protein, ICP4, the protein product of α4. ICP4 is essential for E and L gene expression, and in its absence, infection is unproductive, as it does not proceed past the IE stage [71]. The promoter architecture of E and LL/L genes is less stringent than that of IE genes, with the minimum requirement consisting of only a TATA box [72]. However, E genes commonly contain upstream elements such as GC-rich Sp1 sites [73], and LL/L promoters frequently possess an initiator (Inr) element adjacent to their TSS [74]. In addition, the efficient transactivation of E and LL/L genes absolutely requires ICP4 [75,76]. LL/L genes are distinct from the earlier gene classes, as viral DNA genome replication is essential for the robust expression of these genes [77]. The designation of leaky or true late is dependent on whether DNA replication increases transcription (LL) or is required for transcriptional initiation (L) gene expression [78].

### 3.1. Transcriptional Regulation of the Viral Temporal Cascade

The temporal regulation of viral gene expression is primarily at the transcriptional level, and the protein products of IE genes are the most important in establishing and regulating the temporal cascade. Accordingly, they all (except α47) function primarily to regulate transcription. ICP4 (α4) is an essential trans-activator of E genes but is also a transcriptional repressor of IE genes [71]. ICP0 (α0) is commonly referred to as a promiscuous trans-activator due to its ability to activate the transcription of all kinetic classes of HSV-1 genes [79,80,81]. ICP22 (α22/US1) negatively regulates Pol II processivity on IE genes [82] and has also been linked to the promotion of elongation at later time points of infection [83]. ICP27 (α27/UL54) has multiple functions associated with gene expression, including 3′ processing, splicing [84,85], and mRNA export [86].

A long-standing model proposes that the HSV temporal cascade is controlled through sequential transactivation steps on different viral promoters at different times after infection [66]. It is believed that this occurs due to the differing promoter structures and accessibility of temporal gene classes, leading to the recruitment of specific viral and cellular transcription factors [87]. However, the data supporting this model assessed transcriptional activity through final mRNA/protein output or assays such as Northern blotting, with low sensitivities. These methods do not provide information on nascent mammalian Pol II activity through a gene beginning with initiation, followed by PPP, elongation, and termination [88] (as outlined above). Recent advances in nascent-RNA sequencing technologies have allowed the quantification of Pol II activity at nucleotide resolution and have subsequently revealed new complexities associated with these regulatory steps (reviewed in [8]), including during herpes virus infection [89,90,91].

Precision nuclear run-on sequencing (PRO-Seq) is a nascent RNA sequencing technique that has now been used extensively to study herpes virus transcription [70,82,92,93,94,95,96,97,98]. This involves performing a nuclear run-on of nuclei extracted from infected cells in the presence of biotinylated nucleotides (biotin-NTPs). The incorporation of biotin-NTP’s into the 3′ end of nascent RNA prevents further elongation, allowing for the strand-specific identification of Pol II location at nucleotide resolution [34,99]. Notably, PRO-Seq does not require immunoprecipitation, a limitation of other techniques which report nascent transcripts through the antibody enrichment of Pol II [100].

PRO-Seq has identified an additional step in HSV-1 transcription by revealing that Pol II pauses on HSV-1 genes [93]. In HSV-1, PPP occurs 20–100 nucleotides downstream of the TSS before release into elongation and is presumed to be an important regulatory checkpoint of transcription, much as it is on cellular genes [36]. PPP on the HSV-1 genome occurs on all temporal gene classes, and late viral genes exhibit PPP early in infection, even when they are not being expressed [93]. Overall, these data indicate that the transcriptional regulation of the HSV-1 temporal cascade is more complex than the recruitment of Pol II complexes to different genes at different times after infection.

### 3.2. Transient Immediate–Early Gene-Mediated Repression

Recently, PRO-seq analyses of Pol II activity on the HSV-1 genomes defective for expression of either ICP4, ICP0, or ICP22 revealed unexpectedly increased transcription over that of wild-type viruses in the first 1.5 hpi [94]. This observation indicated that, prior to their role in transcriptional activation, IE gene products first function to repress transcription. This result challenges the previous paradigm of how the temporal cascade establishes and suggests that Pol II/transcription factors are not recruited to specific promoters for the temporal regulation of transcription. Instead, it suggests that Pol II activity is prone to initiate the transcription of all viral genes immediately upon infection. In wild-type viruses, IE genes repress this global transcription before de-repressing different kinetic gene subsets at the appropriate time post-infection. The immediate–early repressive phenomenon has been termed transient immediate–early gene-mediated repression, or TIEMR [70,94].

The specific effect of individual IE genes on TIEMR varies with each gene, but the absence of any component of the TIEMR machinery leads to enhanced transcriptional activity across the entire viral genome within the first 90 min of infection [94]. Notably, this includes increased transcription of the non-IE genes, which are expected to be transcriptionally silent at immediate–early timepoints. This widespread transcription also includes aberrant characteristics such as antisense and intergenic transcription, which are normally precluded by TIEMR [82,94]. Studies using the protein synthesis inhibitor cycloheximide (CHX) indicate that this early repression requires both virion proteins (including ICP4 and ICP0 [101]) and de novo IE protein synthesis, therefore implying roles for non-virion IE proteins such as ICP22. Consistent with the possibility that viral proteins made de novo are important for TIEMR, time-course PRO-Seq studies indicate that even during wild-type infection, non-IE genes are transcribed by active Pol within 30 min of infection, but this activity is quickly repressed within 60 min [70]. Without correct TIEMR, in contrast, activity on these genes continues to increase. A schematic showing the gene expression patterns that occur in IE and non-IE gene subsets, with and without TIEMR, over the first 90 min of infection is shown in Figure 2 (based on data from [70]).

The need for almost immediate transcriptional repression of the HSV-1 genome is not unexpected because viral genomes are essentially transduced plasmids bearing many strong promoters that enter the nucleus free of nucleosomes [102]. Furthermore, viral genes are densely packed together in comparison to the cellular genome, and transcription from genes encoded on opposite strands overlaps. This dictates a high density of promoter elements, including TATA boxes [72], Inr elements [74], and GC-boxes [103], that are recognized by Pol II and other cellular transcription factors, including Sp1 [103] and TBP [104]. HSV-1’s transcriptional competence is supported by the fact that viral VP16, co-introduced into the nucleus alongside the viral genome, is one of the strongest known trans-activators [105]. The transactivating domain (TAD) of VP16 interacts with many transcription factors such as TFIIA, TFIIB, TBP, and TFIIH [105]. It has also been shown that IE genes exhibit high levels of initiation, and that Pol II is highly processive on these genes [70]. In addition, by only 3 hpi, the viral genome harbors 1/3 of all cellular Pol activity [92], highlighting the strong repurposing of Pol II activity to transcribe viral genes.

The concurrent transcription of multiple densely arranged HSV genes would be expected to enter neighboring genes unless this transcription is terminated or precluded. Activity in the absence of TIEMR would likely increase double-stranded RNA (dsRNA) production, a key trigger for the innate immune response [106]. Taken together, these findings indicate that Pol II and associated complexes are rapidly recruited to the viral genome. It is therefore logical that the virus provides a mechanism to ensure the transcriptional repression of later kinetic classes in this highly transcriptionally competent environment, especially in the face of a potentially robust innate immune response.

The loss of ICP4, ICP0, or ICP22 disrupts TIEMR to varying extents [94]. Because each IE protein affects the production of the others, it is difficult to determine whether the role of each is direct or indirect. It is also probable that these IE proteins act together in one or more transcriptional complexes. For example, ICP0 and ICP4 [107] have already been shown to interact with one another, and ICP4 functions in a transcriptional complex containing ICP22 [108]. Nevertheless, all IE proteins function to regulate transcription at immediate–early time points post-infection, which is vital to ensure the correct establishment of the transcriptional cascade. The involvement of these individual IE proteins is discussed in more detail below.

### 3.3. Functions of HSV-1 IE Proteins in Early Transcriptional Regulation

#### 3.3.1. ICP4

ICP4 is a virion protein [101] and a major viral transcriptional regulator that is essential for E gene expression. ICP4 is a phosphoprotein, consisting of an N-terminal activation domain, a DNA-binding domain, and a C-terminal activation domain [109]. Transactivation occurs through ICP4’s ability to directly bind DNA [110] and its subsequent interaction with components of the PIC, including TBP and Mediator [104,111,112]. Chromatin immunoprecipitation (ChIP) studies have indicated that ICP4’s association with many general transcription factors (GTFs) contributes to the highly competent transcriptional environment that occurs on the viral genome during infection [102]. Interestingly, this environment has been shown to exhibit features consistent with liquid–liquid phase separation to form transcriptional condensates [113], the formation of which require ICP4 [114]. The transcriptional condensate model is currently a leading hypothesis explaining how Pol II activity is regulated [115]. This model proposes that Pol II/GTF activity promotes the formation of a distinct liquid-phase droplet with a high concentration of these factors, resulting in a “transcription factory”. ICP4 can also activate the transcription of cellular genes, and has been proposed to reflect its association with NELF, thereby reducing PPP and increasing release to elongation [116].

Chromatin immunoprecipitation followed by deep sequencing (ChIP-Seq) studies have suggested that direct ICP4-mediated Pol II recruitment to viral promoters is key to regulating the temporal cascade [102]. However, PRO-Seq/PRO-Cap experiments revealed that transcriptional activity on non-IE promoters occurs even in the absence of ICP4 [70,94], therefore indicating that ICP4 is not essential to recruit Pol II to these promoters. On the other hand, ICP4 is required for the increased occupancy of active Pol II complexes on early and late genes later in infection. ICP4 may also function indirectly to increase Pol II activity by binding and unfolding DNA at G-rich G-quadruplexes (G4s) on the viral genome [117]. However, the exact mechanism through which ICP4 functions to depress and transactivate viral E genes remains to be elucidated. A schematic displaying ICP4’s known interactions with the eukaryotic transcription machinery to promote transcription is displayed in Figure 3A.

Somewhat paradoxically, ICP4 is also a transcriptional repressor of IE genes. Repression is also associated with ICP4’s DNA-binding activity, occurring at a specific motif in IE promoters, leading to the formation of a repressive tripartite complex containing ICP4, TFIIB, TBP, or TFIID [118,119]. The repression of IE gene initiation is known to occur due to an ICP4 binding site located within 45 bp of the 3′ TATA box, allowing for the formation of the repressive tripartite complex [118,120]. Without ICP4, IE genes are overexpressed, and the cascade does not proceed past IE gene expression. ICP4 is also involved in the repression of cellular transcription, possibly via its association with GTFs and the recruitment of these away from the cellular genome [102]. ICP4’s DNA-binding activity has also been linked to host shut-off, as it has been theorized that ICP4 occupancy can block Pol II recruitment on cellular promoters [116].

PRO-Seq studies indicated that ICP4 is of key importance in TIEMR due to its loss leading to the largest increase in transcriptional activity of all IE proteins [94]. Further analyses using a modified PRO-Seq technique, PRO-Cap, to identify capped RNA initiating at TSSs [99] revealed that ICP4 functions to repress initiation at viral TSSs [70], supporting results from previous studies performed using in vitro transcription and primer extension assays [118]. The use of CHX to prevent protein synthesis during PRO-Seq revealed that virion-associated ICP4 preferentially represses IE promoters, particularly L/S junction spanning transcripts (L/STs) [94], of which ICP4 is already known to be a strong repressor [121,122]. In summary, ICP4 directly represses initiation immediately upon infection at promoters with a high affinity for direct DNA binding; a schematic of this is shown in Figure 3B. It is likely that VP16 transactivation allows for the de-repression of these IE promoters to begin the temporal cascade, and potentially explains why L/ST remains repressed, possibly because it does not contain a TAATGARAT.

**Figure 3 microorganisms-12-00262-f003:**
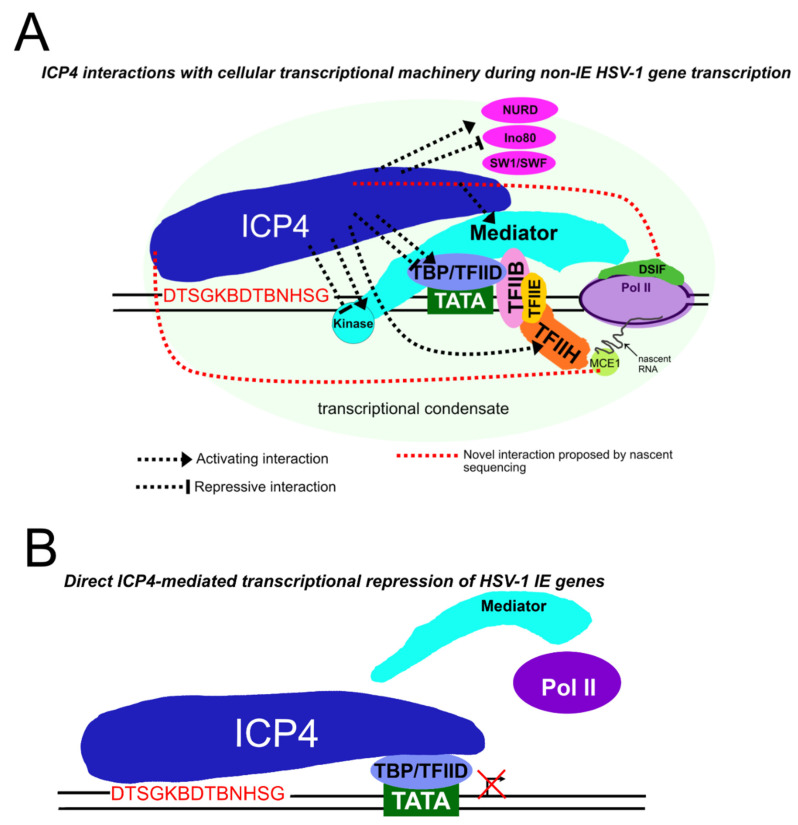
Model of ICP4 function in transcriptional regulation: (**A**) The known interactions of ICP4 and the cellular transcription machinery that are known to occur during non-IE HSV-1 gene transcription. This model assumes that ICP4 binds with low affinity to its consensus motif, DTSGKBDTBNHSG (where D is A, G, T; S is C or G; K is G or T; B is C, G, T; H is A, C, T), identified in [102]. ICP4 interacts with multiple components of the mediator complex, TBP, TFIID, and TFIIH [112]. Other factors such as NURD, Ino80, and SW1/SNF [112] also interact with ICP4 but it is unknown where these are located in this viral chromatin. The recruitment of these factors is believed to aid the formation of transcriptional condensates. Interactions that are presumed to be activating are shown with pointed arrows, and those that are potentially repressive are denoted with blunted arrows. The novel finding identified through nascent sequencing analyses indicating that ICP4 is required for efficient capping and elongation on some genes [70] is denoted with red dotted lines. (**B**) ICP4 directly represses transcription of IE genes via high-affinity binding between its DNA-binding domain and its consensus binding motif in the viral genome. ICP4 then forms a tripartite complex in association with TBP and TFIID. It is assumed that the high affinity for ICP4 to bind at these sites and form a tripartite complex restricts recruitment of Pol II and associated transcription factors [123].

ICP4 also represses the initiation of non-IE genes at immediate–early time points, as these promoters are unexpectedly active in the absence of ICP4 [94]. This occurs even though their promoters do not contain high-affinity ICP4 binding sites, suggesting an alternative repression mechanism. Enhanced transcription in the absence of ICP4 on LL/L genes also included aberrant features: (i) the production of uncapped nascent RNA on some genes, suggesting a role for ICP4 in coupling the capping machinery with the initiation machinery, and (ii) stalling Pol II in some gene bodies, suggesting a facilitating role in elongation under some circumstances [70]. The precise mechanism of these repressive functions is unknown but may be linked to ICP4’s interaction with PIC components, including multiple subunits of the Mediator complex, notably the repressive kinase module [112]. As discussed above, the kinase module regulates transcriptional repression via its interaction with Pol II and the resulting conformational changes [124,125]. Potentially repressive interactions of ICP4 with transcriptional machinery are noted in Figure 3A, alongside the novel potential interactions discovered through nascent sequencing. In addition, ChIP-Seq studies indicated that ICP4 can bind the viral genome with a loose consensus sequence, resulting in it coating the viral genome, hypothesized to form “viral chromatin” in the absence of cellular histones [102]. ICP4 can interact with chromatin remodelers such as SWI/SWF, NURD, and Ino80 [112], and it is possible that these are involved in the formation and maintenance of repressive viral chromatin. The viral genome itself is also associated with multiple cellular proteins at only 1 hpi, including those known to be involved in DNA silencing, such as PML [126] and IFI16 [127].

Taken together, these studies suggest that ICP4 represses transcription through multiple mechanisms. This likely occurs through both direct DNA binding and subsequent interactions with PIC components, and also through involvement in the formation of viral chromatin. The composition of such viral chromatin is likely key in regulating transcriptional repression/de-repression. A remaining intriguing question is how ICP4 switches between roles as a transcriptional repressor or activator.

#### 3.3.2. ICP0

ICP0 is 775 amino acids in length and described as a general trans-activator due to its ability to activate the transcription of all viral kinetic classes. It is classified as a minor virion component [101]. ICP0 is not essential for viral replication, and at high multiplicities of infection (MOIs), ICP0-null mutants grow to equivalent levels to wild-type viruses [128]. However, growth is impaired at low MOIs. The role of ICP0 in transcription is believed to reflect ICP0-dependent orchestration to a cellular state that is more permissive to infection [129]. ICP0 modifies the cellular state through its N-terminal RING-finger domain that mediates interactions with E2 enzymes to provide ICP0 E3 ubiquitin ligase activity [130]. The E3 ubiquitin ligase activity induces the degradation of cellular proteins that would otherwise attempt to silence the viral genome early in infection [131]. This activity is important to ensure the efficient transcription of viral genes within the first 30 min of infection [132].

ICP0 targets components of ND10 bodies, also known as PML nuclear bodies (PML-NBs). During infection with ICP0-null mutants at a low MOI, the viral genome is rapidly entrapped by PML-NBs and replication is restricted [133]. Components of the bodies include PML and sp100, which commonly exhibit high levels of SUMO post-translational modifications. ICP0 contains multiple SUMO-interacting motifs [134], and therefore preferentially induces the proteasomal degradation of SUMOlyated PML and sp100, although the modification is not essential for their degradation [135,136]. ICP0 can also target cellular proteins not associated with PML-NBs but that associate with viral DNA. This targeting results in the degradation of direct transcriptional repressors, such as SLFN5 [137] and IFI16 [138], to enhance viral transcription. The degradation of IFI16 and other proteins such as IRF3 and CBP/p300 [139] reduces the levels of DNA sensors that can trigger the innate immune response [140]. ICP0 is also known to target multiple components of the DNA damage response (DDR), including the DNA-dependent protein kinase (DNA-PK) [141], RNF8, and RNF168 [142]. It is not fully understood why the DDR is targeted by ICP0, but it is believed to enhance the evasion of antiviral silencing mechanisms [143].

ICP0 is not a transcriptional activator in the traditional sense as it neither interacts with Pol II/transcription complexes nor binds DNA directly. It has therefore been proposed as a “DNA template remodeler” rather than a transcriptional activator [131]. Nonetheless, a common theme among the broad range of targets impacted by ICP0 suggests a role in reducing silencing on the viral genome early in infection. It was therefore somewhat surprising that PRO-Seq studies using an ICP0-null mutant detected increased transcriptional activity in the absence of ICP0 [94]. The pattern of increased transcription without ICP0 differed from that of other IE genes implicated in TIEMR: Pol activity remained well regulated, with little antisense or intergenic transcription evident, but E genes were strongly activated at times in which only IE genes should have been transcribed [94]. It is important to note two caveats: (i) these PRO-Seq studies used a high MOI, a condition that helps the virus overcome replication deficits associated with the loss of ICP0 [128], and (ii) ICP0 mutants produce large numbers of defective particles [144], therefore likely introducing high levels of virion components, particularly VP16 and ICP4, that may reduce silencing and enhance procession through the temporal cascade.

We speculate that the altered timing of Pol II and increased activity on certain genes in the absence of ICP0 mutants reflects the indirect and downstream effects caused by TIEMR disruption. For example, the viral template must engage appropriate transcriptional complexes comprising both cellular and viral proteins, likely in a very precise manner, to form viral chromatin (see above). TIEMR may facilitate the formation of viral chromatin by promoting silencing initially to allow appropriate viral and cellular transcription factors to engage the genome in an appropriate manner. It follows that without the correct levels and cooperativity of each IE protein to mediate TIEMR, the formation of the transcriptional environment and subsequent Pol II activity on the viral genome is altered. Transcriptional condensates also potentially facilitate the construction of viral chromatin and interactions between IE proteins. Although ICP4 has been identified as the main requirement for the formation of viral transcriptional condensates [114], all IE proteins are predicted to contain intrinsically disordered regions (IDRs), and such regions are required for the formation of such condensates [113]. Thus, the IE proteins may cooperate to form appropriate condensates, potentially enhancing their functions. Moreover, each IE protein directly influences the expression of the others and affects virion composition [144,145], which potentially changes the milieu of viral proteins present upon the initiation of infection and the assembly of viral chromatin. While these considerations are consistent with previous observations, further studies will be needed to test the role(s) of ICP0 in TIEMR implementation, with a particular emphasis on the nature of viral chromatin.

#### 3.3.3. ICP22

ICP22, encoded by US1, is an IE protein of approximately 68,000 M_r_ that is produced in multiple phosphorylated forms during infection [146,147]. Its expression is required for the efficient establishment of latency in mice [148]. A truncated form of ICP22 is produced from initiation at a downstream methionine within an RNA transcribed from US1.5 that co-terminates with US1 [149,150,151]. Besides its role in viral transcription, ICP22 mediates many functions, including the regulation of cellular gene transcription, the egress of capsids from the nucleus, the activities of cellular cyclin-dependent kinases, and which viral and cellular components are incorporated into the virus particle. These aspects have been reviewed elsewhere [82,145,146,151,152,153,154,155,156].

Much research on ICP22 has focused on its role in altering the CDK9-mediated phosphorylation of the C-terminal domain (CTD) of the RPB1 subunit of Pol II at Ser 2 during infection [157,158,159,160,161,162]. A caveat is that recent data indicates that the CTD Ser2 is more readily phosphorylated by kinases other than CDK9, including cyclin-dependent kinases 12 and 13. The current view is that Ser2 CTD phosphorylation is most important during elongation and termination rather than release from PPP [36,43,57,163,164,165].

ICP22 is itself phosphorylated directly by two viral kinases, US3 and UL13, and can be co-immunoprecipitated with cellular CDK9 [157,158,161,162,166,167]. It is possible that ICP22’s association with any one or all of these kinases at different times post-infection is responsible for the alteration of Pol II phosphorylation during infection [157,161,167]. Research to identify which cyclin-dependent kinase or viral kinase is responsible may be difficult, because it is unknown whether the commonly used CDK9 inhibitors, Flavopiridol (FVP) and DRB, inhibit the viral kinase activity of US3 or UL13 [157,158,160,161,162]. Both of these viral kinases are virion proteins and should be available to the Pol II transcriptional complex before de novo ICP22 expression [168,169,170]. Additionally, FVP, which has been used in experiments with HSV-1, is not CDK9-specific, particularly at high concentrations [160]. At levels > 10 mM, FVP inhibits a wide variety of cellular kinases, including CDK9, CDK7, CDK2, CDK4, CDK1 c-Src, and CDK6 [171,172]. Further research is therefore needed to determine (i) what cellular or viral kinases are present in viral transcriptional complexes, and at what times, (ii) if the commonly used cdk9 inhibitors also inhibit UL13 and US3, (iii) if the viral kinases can phosphorylate cellular targets within viral transcriptional complexes, (iv) if the disruption of kinase activity affects nascent RNA expression at early time points post-infection, and (v) how other, more specific kinase inhibitors affect HSV-1 infection and nascent viral RNA production.

A series of elegant iPOND experiments from DeLuca and colleagues showed that a number of transcriptional elongation components associate with incoming and newly replicated viral DNA [83,126,173,174]. Specifically, iPOND experiments analyzing proteins in association with viral DNA from an ICP22 truncation mutant (n199) identified that the association of CDK9, the FACT complex (Spt16 and SSRP1), the Spt5 subunit of DSIF, and Spt 6 all interact with the viral genome in an ICP22-dependent manner [83,173]. A schematic of these interactions is shown in Figure 4. ICP22 expression is required for the HSV-1-induced changes in phosphorylation on Ser2 in the heptad repeat of the Pol II RPB1 subunit’s CTD [157,158,160,175,176,177,178], although it is not a kinase itself. Therefore, a predominant hypothesis in the field has been that ICP22 expression regulates Pol II after initiation, as these events correlate with Ser2 phosphorylation in the RPB1 subunit [8,33,82,83,93,179].

While ICP22 is not incorporated into viral particles like ICP0 and ICP4 [101,145,180,181], it is produced rapidly de novo, allowing for its detection via immunoblot within 1.5 hpi [94]. Thus, it is likely that ICP22 functions slightly later in TIEMR and after the initial repression dictated by incoming ICP0 and ICP4 [82,94]. Recent PRO-seq and GRO-seq analyses of cells infected with a virus bearing a deletion of the US1/1.5 open reading frames (ΔICP22) revealed a lack of Pol II PPP on key IE genes, including α4 and α0 [82,94]. These results suggest an absence of the machinery required to induce pausing at early times on ΔICP22 IE gene promoters. Possible roles of ICP22 in accentuating this pausing include its action as a negative elongation factor or as a recruiter of cellular factors that inhibit elongation. The latter possibility is consistent with previous iPOND results showing that a truncated form of ICP22 is unable to efficiently enhance the occupancy of DISF (Spt5) on the viral genome [83] (Figure 4).

Additionally, the ΔICP22 mutant is unable to control the rate of Pol elongation throughout the viral genome [82]. These observations suggest that ICP22 slows the rate of transcription on viral IE genes [70,82,94]. This information is potentially consistent with ICP22’s ability to directly bind the FACT complex and the Spt6 elongation factor [83,162] (Figure 4). If this binding slowed elongation, it could facilitate the establishment of virally orchestrated chromatin by ensuring the recruitment of chromatin remodeling factors and histone chaperones to the viral genome. This viral chromatin would then optimize proper mRNA production. Further studies to test these possibilities might include efforts to separate the ICP22 protein domains required to interact with the various components of the pausing and elongation machinery in both epithelial and neuronal cells.

## 4. Summary

Taken together, current PRO-seq and other deep sequencing techniques have revealed an important repression step preceding HSV-1 transcription. This step, which we refer to as TIEMR, ensures the subsequent orderly progression of the viral transcriptional cascade. In the absence of TIEMR, all genes are transcribed at once, with little regard to the sense, anti-sense, or termination of transcription. Such a step may not be surprising given that, when introduced into the nucleus of an infected cell, the HSV genome is essentially a transduced expression plasmid bearing more than 80 strong promoters. While it has been assumed that the repression of late genes at early times was a function of repressive nucleosomes, the PRO-Seq data of IE mutants indicate that functions of IE proteins, especially ICP4 and ICP22, are required to repress all non-IE genes at immediate–early times. We believe it is likely that ICP4 and ICP22 exploit and modify existing cellular mechanisms to repress the expression of viral genes, just as they must exploit other cellular processes to reverse this repression in an orderly manner later in infection. Such hypotheses warrant further exploration, especially regarding the structure and function of viral chromatin, of which ICP0, ICP4, and ICP22 are likely to be important components.

## Figures and Tables

**Figure 1 microorganisms-12-00262-f001:**
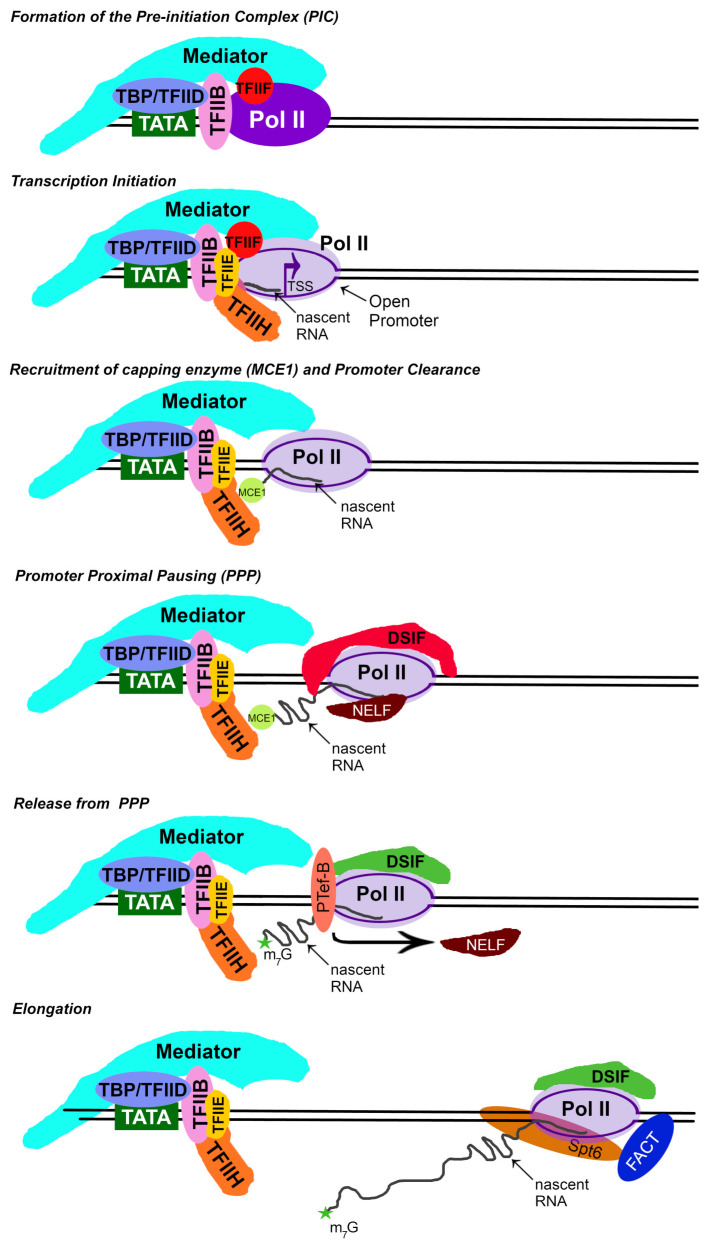
Steps in eukaryotic transcription: (i) TBP/TFIID recognizes TATA box-like sequences in the promoter of a gene. This complex recruits TFIIB to the complex to bend the DNA and recruit Pol II together with TFIIF. Mediator then lies over the complex connecting signals from transcription factors binding upstream of the TFII general transcription factors. Together, this set of proteins is called the pre-initiation complex (PIC). (ii) Transcription is initiated when Mediator recruits the TFIIE and TFIIH factors to the complex. This opens the promoter, loading the coding strand into Pol II to commence transcription. (iii) The capping enzyme (MCE1) is brought to the initiated complex before Pol II clears the promoter in part through TFIIH cyclin-dependent kinase activity. The nascent RNA is captured by MCE1 to facilitate capping. Phosphorylation events driven by TFIIH disassociate Pol II from the PIC. (iv) Once Pol II has cleared the promoter, DSIF and NELF induce promoter proximal pausing. Pausing allows MCE1 to complete the capping process. (v) Positive transcription elongation factor B (P-TEFb) relieves promoter proximal pausing by phosphorylating DSIF, NELF, and Pol II after capping is completed. This releases Pol II from pausing, induces the recruitment of elongation factors, and causes conformational changes within the multi-subunit Pol II. All these actions ultimately increase the rate of transcription. (vi) Pol transcribes the gene body rapidly in the process of elongation, while FACT disassembles the nucleosomes in front of Pol II and Spt6 reassembles them after Pol II passes.

**Figure 2 microorganisms-12-00262-f002:**
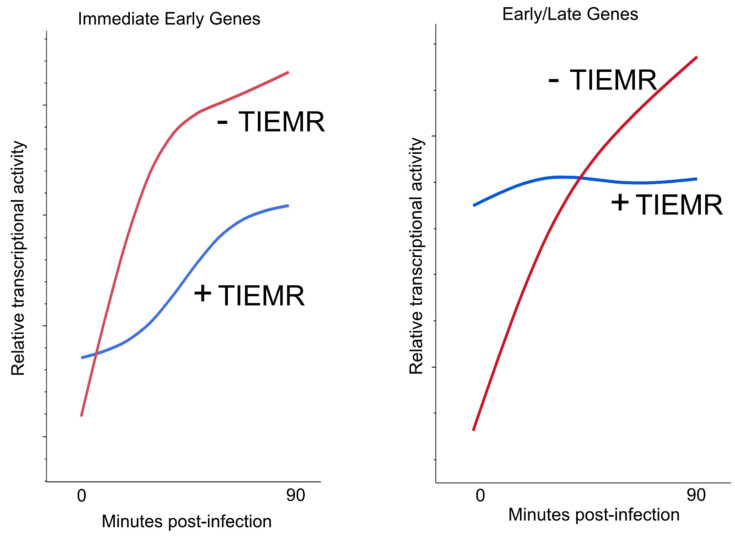
Schematic of transcriptional activity of HSV-1 genes −/+ TIEMR during the first 90 min of infection. Without functioning TIEMR, Pol activity on IE genes increases logarithmically, whereas with TIEMR, activity increases logistically, indicating more controlled expression. Without TIEMR, activity on E/L genes also increases logarithmically in the first 90 min. However, with TIEMR, the genes become rapidly engaged by Pol but remain repressed and do not increase in activity (based on data from [70]).

**Figure 4 microorganisms-12-00262-f004:**
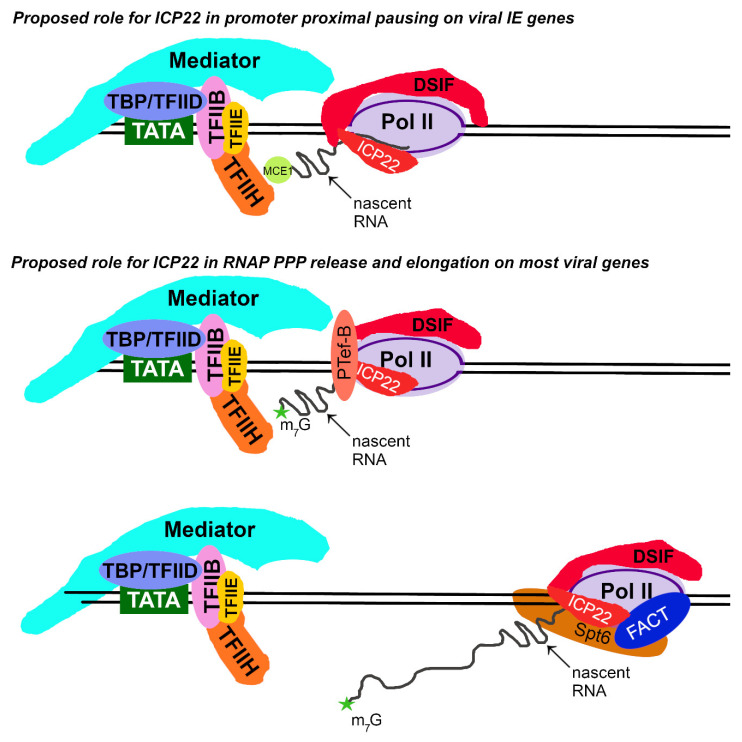
Proposed roles for ICP22 in regulating transcription from viral IE and non-IE genes: Several studies indicate that ICP22 directly or indirectly interacts with CDK9 in the P-TEFb complex, DSIF, Spt6, and the FACT complex [83,157,162,175]. These data combined with PRO-seq and GRO-seq [82,94] suggest that ICP22 interacts with these complexes to induce promoter proximal pausing on viral IE genes and to slow the rate of elongation on other viral genes. These activities likely help to facilitate the orderly and controlled expression of tightly clustered viral genes.

## Data Availability

No data was generated for this manuscript, all information has been gathered from previously published, peer reviewed scientific manuscripts.

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
