# Peer review of "A Revision of Herpes Simplex Virus Type 1 Transcription: First, Repress; Then, Express"

_microorganisms, 2024, doi:10.3390/microorganisms12020262_

Round 1
Reviewer 1 Report
Comments and Suggestions for Authors
The authors have written 5 pages ( total 10 pages) of review article about GENERAL EUKARYOTIC TRANSCRIPTION. The figures are about GENERAL EUKARYOTIC TRANSCRIPTION. How does your review justify HSV and Transcription. You have not shown even a single HSV protein interaction or influencing the HOST transcription. You should be careful in future about such writing.
Author Response
We feel that the first section (now section 2 of revised manuscript) describing eukaryotic transcription is important to include because it defines the cellular machinery with which HSV proteins interact. These interactions are described in section 3 of the revised manuscript.
Reviewer 2 Report
Comments and Suggestions for Authors
I would like to recommend this manuscript for publication after minor revision:
1. Generally speaking, the first part of an academic paper should be the Introduction, which introduces the background and outline of the topic. Therefore, authors are encouraged to supplement this section.
2.The title of the paper is also too colloquial, and authors can add some information to make the title more specific.
3. Usually, few papers display images in subheadings, and authors can move Fig. 1 to the paragraph.
4. The manuscript lacks the corresponding author and their contact information, which is not convenient for readers to contact and discuss in the future. Please indicate the corresponding author and provide additional information.
5. What data is Figure 2 drawn based on? Please provide the data source.
6. Is it appropriate to merge Sections 5-7 into one Section?
Author Response
We appreciate the reviewers comments.
- Generally speaking, the first part of an academic paper should be the Introduction, which introduces the background and outline of the topic. Therefore, authors are encouraged to supplement this section.
- Response, and introduction is now included as section 1.
2.The title of the paper is also too colloquial, and authors can add some information to make the title more specific.
-We feel that the title captures the salient point and should garner reader attention.
- Usually, few papers display images in subheadings, and authors can move Fig. 1 to the paragraph.
Reference to figure 1 is now included in the paragraph as requested.
- The manuscript lacks the corresponding author and their contact information, which is not convenient for readers to contact and discuss in the future. Please indicate the corresponding author and provide additional information.
Corresponding author and email address are now included.
- What data is Figure 2 drawn based on? Please provide the data source.
The published reference containing the data in figure 2 is now included.
- Is it appropriate to merge Sections 5-7 into one Section?
We feel that these topics warrant separate sections, but have placed them in
A subsection of section 2, eukaryotic transcription..
Reviewer 3 Report
Comments and Suggestions for Authors
This is a review article that is timely. It addresses an important question in HSV biology, and that is how does the virus manipulate/utilize the immediate heterochromatinization of the viral genome upon entry by the host cell. Why is this important? It is well documented by many labs that the HSV genome (and other herpesvirus genomes, like HCMV) are immediately heterochromatinized by the host cell upon entry as a defense mechanism. The virus must balance this host defense by its own mechanisms while remaining largely undetected by the immune system at these early times. This review compiles the fact that genomes LIMIT (first) their gene expression at these initial stages of infection before ALLOWING for ample expression and protein production at later times. This is fundamental to HSV biology and important.
The nature of the review is highly original. It is a review so it compiles other work, but the idea that repression is required for expression is original and the authors do an excellent job supporting that with available literature.
Author Response
We thank the reviewer for their kind comments.
Round 2
Reviewer 1 Report
Comments and Suggestions for Authors
The authors have done pretty well however, I had asked to make a figure of HSV proteins and transcription complex based on the literature they provide . However, the authors ignored my comments and kept the same eukaryotic transcription complex figure which does not make any sense when you are discussing HSV and transcription. Other minor comments are in the pdf. Some self citations are added unnecessarily.

Author Response
As suggested, we have added new figures that diagram the interactions indicated in the review.